# Recombinant *Ixodes scapularis* Calreticulin Binds Complement Proteins but Does Not Protect *Borrelia burgdorferi* from Complement Killing

**DOI:** 10.3390/pathogens13070560

**Published:** 2024-07-03

**Authors:** Moiz Ashraf Ansari, Thu-Thuy Nguyen, Klaudia Izabela Kocurek, William Tae Heung Kim, Tae Kwon Kim, Albert Mulenga

**Affiliations:** 1Department of Veterinary Pathobiology, School of Veterinary Medicine and Biomedical Sciences, Texas A&M University, College Station, TX 77843, USA; moiz@tamu.edu (M.A.A.); ttnguyen@cvm.tamu.edu (T.-T.N.); wkim@cvm.tamu.edu (W.T.H.K.); 2Department of Chemistry, Texas A&M University, College Station, TX 77843, USA; kikocurek@tamu.edu; 3Department of Diagnostic Medicine/Pathobiology, College of Veterinary Medicine, Kansas State University, Manhattan, KS 66506, USA; tkim09@vet.k-state.edu

**Keywords:** *Ixodes scapularis*, complement cascade, tick calreticulin, *Borrelia burgdorferi*

## Abstract

*Ixodes scapularis* is a blood-feeding obligate ectoparasite responsible for transmitting the Lyme disease (LD) agent, *Borrelia burgdorferi*. During the feeding process, *I. scapularis* injects *B. burgdorferi* into the host along with its saliva, facilitating the transmission and colonization of the LD agent. Tick calreticulin (CRT) is one of the earliest tick saliva proteins identified and is currently utilized as a biomarker for tick bites. Our recent findings revealed elevated levels of CRT in the saliva proteome of *B. burgdorferi*-infected *I. scapularis* nymphs compared to uninfected ticks. Differential precipitation of proteins (DiffPOP) and LC-MS/MS analyses were used to identify the interactions between *Ixs* (*I. scapularis*) CRT and human plasma proteins and further explore its potential role in shielding *B. burgdorferi* from complement killing. We observed that although yeast-expressed recombinant (r) *Ixs*CRT binds to the C1 complex (C1q, C1r, and C1s), the activator of complement via the classical cascade, it did not inhibit the deposition of the membrane attack complex (MAC) via the classical pathway. Intriguingly, r*Ixs*CRT binds intermediate complement proteins (C3, C5, and C9) and reduces MAC deposition through the lectin pathway. Despite the inhibition of MAC deposition in the lectin pathway, r*Ixs*CRT did not protect a serum-sensitive *B. burgdorferi* strain (B314/pBBE22*Luc*) from complement-induced killing. As *B. burgdorferi* establishes a local dermal infection before disseminating to secondary organs, it is noteworthy that r*Ixs*CRT promotes the replication of *B. burgdorferi* in culture. We hypothesize that r*Ixs*CRT may contribute to the transmission and/or host colonization of *B. burgdorferi* by acting as a decoy activator of complement and by fostering *B. burgdorferi* replication at the transmission site.

## 1. Introduction

The black-legged tick, *Ixodes scapularis*, is a 3-host tick blood-feeding ectoparasite that completes its life cycle (larvae to nymph to adult) over 2 years. This tick is considered endemic to several parts of the USA, including Midwest, Northeast, West, Southeast, and Southern USA, and its distribution is dependent on various ecological factors such as climates, vegetation, landscape, and availability of a host [1,2]. *I. scapularis*, which transmits 7 of 16 human tick-borne disease (TBD) agents in the USA [2], including causative agents of anaplasmosis (*Anaplasma phagocytophilum*), babesiosis (*Babesia microti*), ehrlichiosis (*Ehrlichia chaffeensis*), hard tick relapsing fever (*Borrelia miyamotoi*), and *Powassan encephalitis* virus [3,4,5], it is famously known for the transmission of causative agents of Lyme disease (LD): *Borrelia burgdorferi* [6] and the recently described *B. mayonii* [7,8,9]. LD is among the most important human TBD, with ≈476,000 diagnosed cases nationally per year, and can cost up to USD 1 billion, without including suspected, undiagnosed, and nonacute cases. The accurate economic cost of LD would be much higher if all parameters were included when generating estimates [10]. In the absence of effective vaccines against Lyme disease (LD), killing ticks using acaricides, personal protection measures, and avoiding infected ticks are the only available options to prevent LD. However, it is worth noting that a vaccine is available for tick-borne encephalitis virus (TBEV), highlighting the potential for vaccine development against other tick-borne diseases. However, limitations associated with acaricide use such as tick resistance [11], toxicity [12], cost-effectiveness [13], and environmental contamination have necessitated the development of alternative tick control methods. Immunization against tick feeding has emerged as the most promising alternative tick control method [14,15,16]. The identification and target validation of key tick saliva proteins that facilitate the feeding and transmission of TBD agents by ticks is needed before tick antigen-based vaccines to prevent TBD can be developed. Toward this goal, our lab identified and functionally characterized tick saliva proteins secreted by different stages of ticks and different phases of feeding [17,18,19]. Recently, we found *I. scapularis* tick calreticulin (*Ixs*CRT) among proteins that were highly secreted by *B. burgdorferi*-infected *I. scapularis* nymphs [17]. The objective of this study was to characterize the role of *Ixs*CRT as a tick saliva transmission factor of *B. burgdorferi.*

Tick CRT is among the highly conserved tick saliva proteins; it is found to be expressed universally in all developmental stages and tissues of ticks and has been reported to be a secretory protein in the saliva of multiple tick species including *Amblyomma americanum*, *Rhipicephalus microplus*, and *I. scapularis* [20,21,22]. Except for erythrocytes, CRT has been reported in every cell of higher organisms. CRT shows multiple functions including calcium homeostasis, chaperon activity, lytic activity through T and natural killer (NK) cells, the boosting of the phagocytosis of apoptotic cells, and the suppression of tumor growth [23,24]. Similarly, tick CRT is also a calcium ion binding protein and it is believed that since calcium ion is an important factor for blood clotting, ticks might secrete this protein to prevent blood clotting [22]. This concept was explored in another study that showed that *Haemonchus contortus* CRT prevented blood clotting by binding to Ca^2+^ and sequestering blood clotting factors, prothrombin factor X, and C-reactive proteins [25,26].

Several studies reported elevated antibodies for tick CRT following tick bites and, consequently, tick CRT serves as a biomarker for tick bites [20,27,28]. However, the role(s) of tick CRT in feeding and pathogen transmission is not fully known. Functional analyses of CRT in other parasites including *Trypanosoma cruzi*, *Haemonchus contortus*, *Entamoeba histolytica*, and *Trypanosoma carassii* have suggested that parasite CRT blocks the complement cascade by binding with the complement component C1q [23,25,29,30]. In our lab, we previously reported that recombinant *Amblyomma americanum* tick CRT bound C1q, a member of the C1 complex, but did not inhibit the deposition of the terminal complex of the complement cascade [31,32]. Similarly, in this study, we report that yeast-expressed recombinant *I. scapularis* CRT (r*Ixs*CRT) binds to complement C1 complex proteins (C1q, C1r, and C1s). Although the C1 complex activated the complement system via the classical pathway, r*Ixs*CRT binding of the C1 complex did not inhibit the deposition of the membrane attack complex (MAC) in the classical pathway. Interestingly, r*Ixs*CRT also bound intermediate complete system factors (C3, C5, and C9) and significantly reduced MAC deposition via the lectin pathway. Importantly, while r*Ixs*CRT did not rescue *B. burgdorferi* from complement killing, it promoted the growth of *B. burgdorferi* in culture. Our data suggest that *B. burgdorferi* stimulates ticks to secrete high amounts of CRT, which could promote its growth at the tick-feeding site.

## 2. Material and Methods

### 2.1. Expression and Affinity Purification of Recombinant I. scapularis Calreticulin (rIxsCRT)

The expression of r*Ixs*CRT was analyzed using the pPICZαA and *Pichia pastoris* yeast expression system as previously described [32]. Synthesis of the r*Ixs*CRT expression plasmid was outsourced (Biomatik, Wilmington, DE, USA). The mature protein-coding domain was based on the *I. scapularis* CRT sequence in GenBank (accession #AY690335.1), with the inclusion of a C-terminus hexa-histidine fusion tag, and was cloned between the *EcoR*I and *Not*I sites in the pPICZαA plasmid. The r*Ixs*CRT expression plasmid was used to transform *P. pastoris* (X-33), as described [33]. Transformed colonies were selected on Yeast Extract Peptone Dextrose Medium with Sorbitol (YPDS) agar plates with zeocin (100 μg/mL, Invivogen, San Diego, CA, USA) incubated at 28 °C. Positive transformants (confirmed by PCR check) were inoculated in Buffered Glycerol-complex Medium (BMGY) and grown overnight at 28 °C with shaking (230–250 rpm). Subsequently, the cells were used to inoculate Buffered Methanol-complex Medium (BMMY) to *A*_600_ of 1, after which, protein expression was induced by adding methanol up to a 1% final concentration every 24 h. Pilot expression showed that expression levels of r*Ixs*CRT peaked on day 3. Thus, for large-scale 1 L cultures, we cultured yeast for up to three days. The pPICZαA plasmid secreted the recombinant protein into media, and, thus, r*Ixs*CRT was salted out using ammonium sulfate saturation, as described [32]. Precipitated r*Ixs*CRT was dialyzed against column-binding buffer (1 M NaCl, 0.4 M Tris, 0.2 M imidazole, pH 7.4) and was affinity-purified using HiTrap chelating HP column, as per the manufacturer’s instructions (Cytiva, Marlborough, MA, USA). Expression and affinity purification of r*Ixs*CRT was confirmed by SDS-PAGE and visualized by silver staining and Western blotting analysis, using antibodies against the C-terminus hexa-histidine fusion tag (Thermo Fisher Scientific, Waltham, MA, USA). The affinity-purified r*Ixs*CRT was dialyzed against 10 mM HEPES buffer, pH 7.4; 1× PBS, pH 7.4; or normal saline based on appropriate assays described below and stored at −80 °C until use.

### 2.2. Differential Precipitation (DIffPOP) of rIxsCRT and Human Plasma Proteins

To identify plasma proteins that might interact with tick saliva CRT, protein-to-protein interaction using differential precipitation was conducted as published [34,35]. Briefly, a reaction volume of 150 µL was prepared by preincubating r*Ixs*CRT (10 µg) with 10% normal human serum (NHS, Sigma-Aldrich, St. Louis, MO, USA) for 90 min at 37 °C. NHS only and r*Ixs*CRT only were included as negative controls. After incubation, 100 µL of stabilizing buffer (Phosphoprotein Kit, Buffer A, Takara Bio Company, San Jose, CA, USA) was added to the reaction. Subsequently, to fractionate, an escalating amount of precipitation buffer (90% methanol/1% acetic acid) starting at 3.75 µL (or 0.9% of the reaction volume) in a fraction of 1 to 317 µL (or 79.25% of the reaction) in fraction 10 was added to the reaction mixture (Appendix A), followed by vortexing to mix. The reaction was incubated at RT (room temperature) for 5 min and centrifuged at 13,000× *g* for 10 min at 4 °C. The supernatant was carefully transferred to another tube without disturbing the pellet and then transferred to another 1.5 mL centrifuge tube; next, the precipitation process was repeated 9 more times. The collected pellets (*n* = 10) were washed with ice-cold acetone, air-dried, and dissolved in PBS and considered as fractions 1–10. The plasma proteins that interacted with r*Ixs*CRT were expected to co-precipitate with r*Ixs*CRT. Subsequently, each fraction was resolved on 10% SDS-PAGE gel, silver-stained, and subjected to the standard Western blot analysis using antibodies for the hexa-histidine tag to track r*Ixs*CRT fractionation. To identify plasma proteins that interacted with r*Ixs*CRT, fractions were subjected to LC-MS/MS analysis.

### 2.3. LC-MS/MS Analysis

The samples for LC-MS/MS analysis were prepared by the in-solution protein digestion method. In this method, 10 µg of differentially fractionated proteins were acetone-precipitated, reconstituted in 100 µL of digestion buffer (50 mM ammonium bicarbonate in water, pH 8.5), and reduced with DTT (2 mM). The solution was incubated at 60 °C for 30 min and then alkylated with 10 mM of iodoacetamide and incubated in the dark for 30 min at 60 °C. Subsequently, proteins were digested overnight at 37 °C with trypsin at a 50:1 ratio (protein-to-trypsin) and submitted for LC-MS/MS analysis. LC-MS/MS analysis was performed on the Orbitrap Fusion tribrid mass spectrometer (Thermo Fisher Scientific, Bremen, Germany) equipped with a Dionex UltiMate 3000 reverse-phase nano-UHPLC system (Thermo Fisher Scientific, Bremen, Germany). A total of 1 µL of each sample was injected into the device and separated by a 150 × 0.075 mm C18 column (Phenomenex bioZen XB-C18, 2.6 µm particle size, Phenomenex, Torrance, CA, USA) at a flow rate of 0.400 µL/min. The total duration of the method was 60 min, with the gradient as follows: equilibration at 2% B (98% acetonitrile, 2% water, 0.1% formic acid), ramp to 45% B at 37 min, ramp to 90% B at 40 min and hold until 46 min, and ramp down to 2% B at 47 min and hold at 2% B until the end of the run at 60 min.

Eluent was introduced into the Fusion mass spectrometer by nano-ESI at a static voltage of 2450 V, with a transfer capillary temperature of 275 °C. Mass spectrometry data were acquired in positive mode at a resolution of 120,000 (at *m*/*z* 200) in the *m*/*z* range of 400–1600. The RF lens was set to 60%. The maximum injection time was 100 ms. Scans were acquired in top-speed mode with the cycle duration set to 3 s. The intensity threshold for precursors of interest was 5.0 × 10^3^. Charge states 1–6 were considered. Dynamic exclusion was set to 60 s with a mass tolerance of 10 ppm. MS/MS data were acquired by HCD at a fixed collision energy of 28% with a precursor ion isolation window of 1.6 *m*/*z*; fragments were detected in the ion trap at a rapid scan rate. Proteome Discoverer 2.4 software (Thermo Fisher Scientific, Waltham, MA, USA) was used to analyze the MS/MS spectra of peptide ions to search against the publicly available NCBI protein database (www.ncbi.nlm.nih.gov, accessed 13 March 2024). Plasma protein and r*Ixs*CRT interactions were confirmed if the normalized spectral abundance factor was significantly different between control (plasma only) and treatment (plasma and r*Ixs*CRT) samples.

Subsequently, the pathways that were represented by interactions between r*Ixs*CRT and plasma proteins were revealed in the Reactome database (http://reactome.org) [36]. Accession numbers of plasma protein interactors with r*Ixs*CRT were loaded onto the Reactome database server and highly significant pathways were reported.

### 2.4. Pull-Down Assays to Validate Interactions between rIxsCRT and Human Complement Proteins

Prompted by DiffPOP and LC-MS/MS analyses of interactions between complement proteins and r*Ixs*CRT, Dynabeads™ His-Tag Isolation and Pulldown magnetic beads (Thermo Fisher Scientific, Waltham, MA, USA) were used to perform a pull-down assay to validate the interactions. The experiment was performed according to the manufacturer’s guidelines, with minor modifications. Briefly, 50 µL (2 mg) of magnetic Dynabeads™ was transferred to a 1.5 mL microcentrifuge tube and placed on a magnetic stand for 2 min. The supernatant was discarded. Purified r*Ixs*CRT (40 µg) was added to the beads and briefly vortexed to mix. The mixture was incubated on a roller at RT for 2 h. The tube was placed on the magnet for 2 min and the supernatant was discarded. To remove nonspecific interactions, the beads were washed 4 times with the washing buffer (100 mM sodium phosphate (pH 8), 600 mM NaCl, 0.02% Tween-20). The test proteins, human complement serum (HCS; Innovative Research, Inc., Novi, MI, USA) were diluted to 10% in the pull-down buffer (6.5 mM sodium phosphate (pH 7.4), 140 mM NaCl, 0.02% Tween-20), added to the bead–bait complex, and incubated on a roller for 2 h at RT. Then, the mixture was placed on the magnetic stand for 2 min and the supernatant was discarded. The beads were washed 4 times with the wash buffer (100 mM sodium phosphate (pH 8.0), 600 mM NaCl, 0.02% Tween-20) and test protein complexes were eluted by incubating the beads with elution buffer (300 mM Imidazole, 50 mM sodium phosphate (pH 8.0), 300 mM NaCl, 0.01% Tween-20) for 5 min at RT. Subsequently, eluted proteins were subjected to standard SDS-PAGE and Western blot analysis using antibodies for specific complement proteins: C1q (Thermo Fisher Scientific, Waltham, MA, USA), C1r (Thermo Fisher Scientific, Waltham, MA, USA), and C1s (Sino Biological, Inc., Houston, TX, USA), Activated C3 Antibody (I3/15) (Santa Cruz Biotechnology, Inc., Dallas, TX, USA), inactive C3 (Complement Technology, Inc., Tyler, TX, USA), C5 (Complement Technology, Inc., Tyler, TX, USA), and C9 (Complement Technology, Inc., Tyler, TX, USA), as well as the C5b-9 complex or membrane attack complex: MAC (Santa Cruz Biotechnology, Inc., Dallas, TX, USA). For controls, r*Ixs*CRT-only and HCS-only samples were processed.

### 2.5. ELISA Analysis to Validate the rIxsCRT Binding of Complement Proteins

Preliminary differential precipitation and pull-down assays showed that r*Ixs*CRT likely interacted with complement proteins. To further substantiate these interactions, conventional ELISA was used. High Binding plates (Thermo Fisher Scientific, Waltham, MA, USA) were coated with r*Ixs*CRT (250 ng) in ELISA coating buffer (carbonate–bicarbonate buffer, pH 9.6) overnight at 4 °C. Subsequently, wells were washed 3 times with phosphate buffer saline with Tween 20 (PBST) and the non-specific binding sites were blocked using blocking buffer (5% skim milk dissolved in PBST) for 2 h at RT. Following washing, 10% normal human serum (NHS) was added to the wells and incubated for 2 h at RT. The coated plate was washed with PBST and wells were incubated for 3 h with respective primary polyclonal antibodies (Anti-C1q, Anti-C1r, Anti-C1s) and anti-sera (Anti-C3, Anti-C5, and Anti-C9) for complement proteins at a 1:5000 dilution. Wells were washed 3 times with PBST and incubated with respective HRP-conjugated secondary antibodies for 1 h (Anti-Rabbit for C1q, C1r, and C1s and anti-Goat for C3, C5, and C9). Wells were again washed 3 times with PBST, and bound peroxidase activity was observed by adding 1-Step Ultra TMB-ELISA substrate (Thermo Fisher Scientific, Waltham, MA, USA). The reaction was stopped using 2 M sulfuric acid. The intensity of color development was quantified by measuring the intensity at 450 nm using a microplate reader (Biotek Instrument Inc., Winooski, VT, USA).

### 2.6. Effect of rIxsCRT on Complement Activation

The effects of r*Ixs*CRT of complement activation were assayed using the Wieslab^®^ Complement System Screen (Svar Life Science AB, Malmo, Sweden), according to the manufacturer’s recommendation and as previously published [35,37]. This kit evaluates effects on membrane attack complex (MAC) deposition via each of the three complement system activation pathways: classical, lectin, and alternative. To test the effect of the r*Ixs*CRT on the complement pathways, NHS (provided with the kit) was diluted according to the manufacturer’s instructions and incubated at RT for 15 min. Briefly, r*Ixs*CRT (4 μM) was added to the NHS and incubated at 37 °C for 30 min before adding the samples to the wells of the Wieslab^®^ plates. The 100 μL of r*Ixs*CRT and NHS samples (provided in the kit) were then added to the wells of the Wieslab^®^ plates along with a positive control (human serum, provided with the kit); negative controls, provided with the kit; and a blank, (diluent only) in duplicates and incubated at 37 °C for 1 h. Finally, the absorbance was read at 405 nm on a Biotek Synergy H1 microplate reader (Biotek Instrument Inc., Winooski, VT, USA). The MAC deposition rate was calculated as follows: (Sample − NC)/(PC − NC) × 100, where NC is the negative control and PC is the positive control.

### 2.7. Complement Sensitivity Assay

Our preliminary findings showed that r*Ixs*CRT dose-dependently reduced the deposition of the MAC by the mannose-binding lectin pathway, so we were interested in studying its effect on rescuing the complement-sensitive *B. burgdorferi* spirochete. The complement-sensitive *B. burgdorferi* (B314/pBBE22*Luc*) and resistant (B314/pCD100) strains [38] were a kind gift from Jon. T. Skare laboratory (Texas A&M University Health Science Center, College Station, TX, USA). These two strains were genetically modified from the serum-sensitive strain B314, which lacks most linear plasmids, using the shuttle vector pBBE22 [38]. The pBBE22luc expresses luciferase, while the pCD100 expresses luciferase and BBK32, which inhibits the classical pathway of the complement system. Both strains were maintained in BSK-II media supplemented with 6% rabbit serum at 32 °C with 1% CO_2_ for this assay. A total of 15 µL of normal human serum (NHS) (Complement Technology, Inc., Tyler, TX, USA) was pre-incubated with serial dilutions of r*Ixs*CRT (0.25, 0.5, 0.75, and 1 µM) at 37 °C for 30 min prior to the addition of 85 µL of *B. burgdorferi* B314/pBBE22*luc* (10^6^ cells/mL) and inoculated in a bio-shaker at 32 °C with shaking at 100 rpm; PBS (phosphate buffered saline) was used as a blank. NHS with *B. burgdorferi* B314/pBBE22*luc* and B314/pPCD100 were included as negative and positive controls, respectively. The survival of spirochetes was assessed at 1.5 h, 2 h, 2.5 h, and 3 h post-incubation. Spirochetes were counted from randomly chosen fields (10–15 fields) under a dark-field microscope. Spirochete viability was arbitrated based on cell mobility, membrane integrity, and cell lysis, as described elsewhere [38]. Spirochete survival rates were calculated from 3 biological replicates. Heat-inactivated NHS (hiNHS) was used as the no-complement-activity control.

### 2.8. Blood Recalcification Time Assay

To assess the effect of r*Ixs*CRT on blood clotting, the blood recalcification time was calculated as described elsewhere [39,40]. In this assay, normal blood plasma, Pacific Hemostasis™ Coagulation Reference Plasma (Thermo Fisher Scientific, Waltham, MA, USA), was used to measure the recalcification time in the presence of r*Ixs*CRT. In this assay, 50 µL of plasma was incubated with tris-buffer (20 mM Tris-HCl, pH 7.5) and r*Ixs*CRT (2.6 mM) at room temperature for 30 min. Simultaneously, 150 mM CaCl_2_ was incubated at 37 °C for 30 min. After incubation, plasma with r*Ixs*CRT (140 µL total volume) along with prewarmed CaCl_2_ (10 µL) was placed in triplicate (for each sample) in a microwell plate and the absorbance was measured at a 650 nm wavelength for 30 min at 55 s intervals. The sample without r*Ixs*CRT was used as a positive control and tris-buffer only was used as a blank.

### 2.9. Assessing the Effect of rIxsCRT on B. burgdorferi Growth under In Vitro Conditions

To gauge insight into the effect of *Ixs*CRT on *B. burgdorferi*, low-passaged (<6 passages) *B. burgdorferi* sensu stricto B31 (strain MSK5) [41] was co-cultured with r*Ixs*CRT (2.6 mM final concentration) in BSK-II medium (Sigma-Aldrich, St. Louis, MO, USA) supplemented with 6% rabbit serum. *B. burgdorferi* without r*Ixs*CRT was used as a control. Spirochetes were monitored each day under dark-field microscopy to observe any morphological changes. Using the Petroff-Hausser Hemacytometer (Hausser Scientific Co., Horsham, PA, USA), the starting concentration of *B. burgdorferi* was adjusted to 10^5^ cells/mL. From each 5 mL culture tube, 5 μL of BSK-II medium was pipetted onto a Petroff-Hausser Hemacytometer (Hausser Scientific Co., Horsham, PA, USA) and the cells were counted for all cells. The number of spirochetes was determined by using the following formula: Spirochetes/mL = Average of spirochetes counted in all wells × Dilution factor × 50,000 (50,000 = 50 (cell depth 1/50) × 1000 (1000 cubic mm = 1 milliliter).

To quantify *B. burgdorferi*, 200 µL of cells were aliquoted from the culture on days 0, 2, 5, and 7. Genomic DNA (gDNA) for each aliquot was extracted by using the DNeasy Kit, according to the manufacturer’s instructions (Qiagens, Germantown, MD, USA). The DNA concentration was determined by Nanodrop (Thermo Fisher Scientific, Waltham, MA, USA). In triplicate, gDNA (500 ng) was mixed with 300 nM *Fla*B qPCR primers (sense TCTTTTCTCTGGTGAGGGAGCT, anti-sense TCCTTCCTGTTGAACACCCTCT) and the PowerUp SYBR^®^ Green Supermix (Thermo Fisher Scientific, Waltham, MA, USA) at a 20 µL reaction volume and subjected to cycling on the iQ5 Multicolor Real-Time PCR Detection System (Bio-Rad, Hercules, CA, USA). *B. burgdorferi* was quantified by comparative analysis using BioRad CFX Maestro 3.2v software, which analyzed the Ct values between the standards (10^1^ to 10^8^ cells/mL *B. burgdorferi*) and the test cultures (without treatment and r*Ixs*CRT treated with *B. burgdorferi*). The *Borrelia* count was calculated by plotting the standard curves of Ct values versus the log of copy numbers included in each PCR run [42].

### 2.10. Identifying the Reaction Intensity of rIxsCRT against B. burgdorferi-Infected Rabbit IgG

To ascertain the reactivity of r*Ixs*CRT towards *B. burgdorferi*-infected rabbit IgG [17], both ELISA and Western blot assays were conducted. The ELISA and Western blot were performed following similar steps as those mentioned above. For both the assays, 40 µg/mL of pre-infested (PI) *B. burgdorferi*-infected and uninfected rabbit IgG was used as the primary antibody and anti-Rabbit IgG-HRP conjugated (Southern Biotech, Birmingham, AL, USA) was used as the secondary antibody (1:10^4^ dilution). The primary antibody used in these assays was purified from the rabbit serum, which was infested once with *B. burgdorferi-*infected and uninfected *I. scapularis*. The IgG was purified using the Protein A/G column (Cytiva, Marlborough, MA, USA), following the company’s instructions.

### 2.11. Statistical Analysis

To evaluate statistical significance, the data were analyzed using GraphPad Prism 9 software (GraphPad Software Inc., La Jolla, CA, USA), representing the mean ± SEM (standard error of the mean), with statistical significance (*p* < 0.05) detected by using the non-parametric Student’s *t*-test and two-tailed ANOVA with 95% confidence intervals.

## 3. Results

### 3.1. Differential Precipitation of Proteins (DiffPOP) and LC-MS/MS Analyses Reveal Multiple Interactions between rIxsCRT Human and Plasma Proteins

Recombinant (r) *Ixs*CRT was successfully expressed in *P. pastoris* (Figure 1). Affinity-purified r*Ixs*CRT migrated at the expected molecular weight size (47.61 kDa), as shown by silver staining (Figure 1A) and Western blotting analyses using antibodies for the histidine fusion tag (Figure 1B).

We previously reported that although *A. americanum* tick CRT bound the C1q complement protein, it did not inhibit the activation of the classical complement cascade [32]. To expand on these findings, we employed a DiffPOP (Figure 2) assay [34]. In this assay, plasma proteins that interacted with r*Ixs*CRT co-precipitated with r*Ix*CRT, and Western blotting analysis using the antibody for the histidine tag identified co-precipitating fractions 1–8 (Figure 2C). These fractions were subjected to LC-MS/MS analysis and divided into groups as pooled lanes 1–5 (hereafter called group A), lane 6 (hereafter called group B), and pooled lanes 7 and 8 (hereafter called group C) to identify the plasma proteins that interacted with r*Ixs*CRT.

Normalized abundance values were used to construct a volcano plot (Figure 3) as a graphical representation of differential co-precipitations of human proteins without or with r*Ixs*CRT. Analysis of abundance values identified 1074, 1617, and 1936 unique proteins that interacted with r*Ixs*CRT in groups A–C, respectively (Appendix A). To gain further insights, plasma proteins that were detected at least 1.5-fold higher in r*Ixs*CRT and plasma reactions compared to plasma-only controls were subjected to reactome analysis (Table 1 and Appendix A).

Group A (pooled fractions 1–5) proteins mapped to nine pathways (Table 1) that included three complement function pathways, the initial triggering of the complement cascade, the regulation of the complement cascade, and the terminal pathway of complement (Table 1). Group A proteins were identified as defective ABCA12 (ATP-binding cassette sub-family A member 12), which causes ARCI4B (autosomal recessive congenital ichthyosis 4B) and is associated with reduced protease regulation and skin-barrier dysfunction [43,44,45,46], and defective ABCC2 (ATP-binding cassette subfamily 2), which causes DJS (Dubin–Johnson syndrome) and is associated with hyperbilirubinemia [47]. Other group A proteins mapped to pathways of defective factor XII and defective SERPING1, which cause hereditary angioedema [48], CD22 (receptor predominantly restricted to B cells)-mediated BCR (B-cell receptors) regulation, and the recycling of bile acids and salts [49].

Group B (fraction 6) proteins mapped to 12 pathways (Table 1) that included the terminal pathway of complement, the regulation of the complement cascade, and the complement cascade. The other pathways associated with group B proteins included the response to elevated platelet cytosolic Ca^2+^, the regulation of Insulin-like Growth Factor (IGF) transport and uptake by Insulin-like Growth Factor Binding Proteins (IGFBPs), platelet degranulation, the dissolution of fibrin clot, post-translational protein phosphorylation, semaphorin interactions, sema4D-induced cell migration and growth-cone collapse, RHO GTPases activate CIT, and the common pathway of fibrin clot formation.

Finally, group C (pooled fractions 7 and 8) proteins mapped to 17 pathways (Table 1) associated with signaling, the extracellular matrix (ECM), and cell adhesion functions. The signaling-related pathways included the NGF (nerve growth factor)-activated Trk (neurotrophin receptor) and netrin-1 signaling, which leads to the release of intracellular calcium [50] and signaling by PDGF (*platelet*-derived growth factor), which is associated with the cellular immune response [51]. The ECM-related pathways included multiple collagen-related functions including collagen chain trimerization, biosynthesis, formation, assembly, crosslinking, and degradation. The other ECM functions that mapped group C proteins included anchoring fibril formation, the degradation of ECM, and laminin interactions.

### 3.2. rIxsCRT and Complement Protein Interactions Validated by Western Blotting and ELISA Analyses

Relative abundance analysis revealed that complement proteins co-precipitated at high levels in the presence of r*Ixs*CRT, as revealed by normalized spectral abundance (Figure 4A,B). To validate these findings, all DiffPOP fractions (Figure 2) were subjected to Western blotting analysis using antibodies to complement proteins including C1q, C1r, C1s, C3 (inactive and activated), C5, C9, and C5b-9 (or the membrane attack complex). This analysis confirmed our differential precipitation and LC-MS/MS analysis data (Figure 3 and Figure 4), as indicated by the high protein band intensity of complement proteins that co-precipitated with r*Ixs*CRT compared to NHS only (red arrowhead marked in Figure 5A–L). Our data also showed that r*Ixs*CRT bound all three subunits of the complement complex 1 (C1q, C1r, and C1s) along with C3, C5, and C9 (Figure 5A–L).

To confirm the binding of complement proteins to r*Ixs*CRT, we employed Dynabeads™ His-Tag magnetic beads for the pull-down of r*Ixs*CRT and plasma protein complexes (Figure 6). Subsequently, these complexes underwent the standard Western blotting analysis using antibodies similar to those utilized in Figure 5. Our rigorous analysis unequivocally validated the association of complement C1 complex proteins (C1q, C1r, and C1s), as well as intermediate complement proteins C3, C5, and C9, with r*Ixs*CRT. Specifically, distinct signals were detected in the r*Ixs*CRT and NHS pull-down samples while being conspicuously absent from the NHS-only control (Figure 6). These findings provide robust evidence of the direct interaction between r*Ixs*CRT and key components of the host complement system, further elucidating the molecular mechanisms underlying tick–host interactions.

To further validate the differential fractionation and pull-down data (Figure 5 and Figure 6), conventional ELISA was used to confirm the r*Ixs*CRT binding of complement proteins. We coated ELISA plates with r*Ixs*CRT and then incubated them with NHS. Uncoated wells were used as a negative control. As shown, we observed significantly high antibody binding in wells that were coated with r*Ixs*CRT compared to uncoated control wells. Our data confirmed that r*Ixs*CRT bound the C1 complex proteins (C1q, C1r, and C1s) and intermediate complement proteins C3 and C5 but not activated C3 or C5b-9 (or the membrane attack complex) (Figure 7).

### 3.3. rIxsCRT Moderately Inhibits Membrane Attack Complex (MAC) Deposition via the Lectin Complement Cascade but Does Not Protect B. burgdorferi from Complement Killing

Prompted by the findings showing that r*Ixs*CRT bound complement proteins, we investigated the effect of this protein on the activation of the complement system using the WIESLAB^®^ complement system kit, which allowed for an independent assessment of the three complement activation pathways. Our data showed that r*Ixs*CRT inhibited the deposition of the complement membrane attack complex (MAC) by 40% via the mannan-binding lectin complement activation pathway but not via the classical and alternative complement activation pathways (Figure 8). We next evaluated if r*Ixs*CRT could protect the complement-sensitive *B. burgdorferi* strain (B314/pBBE22luc) from killing by complement. Our data showed that pre-incubating serum with r*Ixs*CRT did not protect the complement-sensitive strain from killing by complement (Figure 9).

Prompted by findings in our differential precipitation analysis that r*Ixs*CRT interacted with blood clotting factor XII and fibrin clot formation pathways, we investigated the effect of this protein on blood clotting using the recalcification time assay. This analysis showed that r*Ixs*CRT did not affect blood clotting (Appendix A).

### 3.4. IxsCRT Promotes Borrelia burgdorferi Growth in Culture

Next, we investigated the effects of r*Ixs*CRT on the growth of *B*. *burgdorferi* in culture. In this study, we observed that the *B. burgdorferi* (10^5^ cells per mL) grown with r*Ixs*CRT showed significant elevation in the growth rate of *B. burgdorferi* (Figure 10). The spirochetes were quantified by the Petroff-Hausser chamber and processed for genomic DNA extraction for FlaB gene quantification by qPCR. Both of these analyses showed that r*Ixs*CRT promoted *B. burgdorferi* growth in culture at its log phase (Figure 10A,B).

### 3.5. B. burgdorferi-Infected I. scapularis Feeding Elicits High IgG Levels for rIxsCRT in Rabbits

In an earlier study, we identified r*Ixs*CRT as one of the tick saliva proteins that were abundantly secreted by *B. burgdorferi*-infected *I. scapularis* nymphs [19]. Consistent with the dynamic secretion pattern of the r*Ixs*CRT protein, rabbits that were fed upon by *B. burgdorferi*-infected ticks exhibited significantly higher levels of IgG antibodies reacting to r*Ixs*CRT compared to rabbits fed upon by uninfected ticks (Figure 11A–D).

## 4. Discussion

Calreticulin (CRT) is a well-known protein that was originally identified as a calcium ion-binding protein localized to the sarcoplasmic reticulum in rabbits [52]. It was originally believed to be retained in the endoplasmic reticulum and function as a molecular chaperon and for Ca^2+^ homeostasis [53]. However, in ixodid ticks, it was surprisingly found to lack the endoplasmic reticulum (ER) retention signal (KDEL) [22] and has been confirmed to be injected into the host during tick feeding [19]. It is currently used as the biomarker for tick bites [20]. The findings of this study elucidated the role of CRT in tick infestation dynamics. We demonstrated that rabbits infested only once against both uninfected or *B. burgdorferi*-infected *I. scapularis* nymph feeding exhibited elevated antibody titers against recombinant *I. scapularis* calreticulin (r*Ixs*CRT), affirming its potential as a reliable marker for tick bites. These findings not only underscore the multifaceted functions of CRT but also highlight its practical application in diagnosing tick infestations.

Studies in mammals have reported a plethora of CRT functions including calcium storage, cell proliferation, ion binding, apoptosis, and cell differentiation [53]. Likewise, another parasite CRT was linked to parasite evasion of the complement through the sequestration of complement proteins and blocking the blood clotting system through the binding of blood clotting factor Xa [54]. While further validation studies are needed, our differential precipitation and LC-MS/MS analysis data showing that r*Ixs*CRT interacted with multiple pathways indicate that tick CRT is likely involved in regulating several functions at the tick-feeding site. Consistent with published studies that reported tick and other parasite CRT interactions with complement and blood clotting pathway proteins [55,56], r*Ixs*CRT differentially co-precipitated with proteins associated with these two pathways. Since the tick must evade both complement and blood-clotting host defenses to successfully feed and transmit tick-borne disease pathogens [57], the findings of this study suggest a role of *Ixs*CRT in mediating tick evasion of host defense functions.

Evasion of the complement system by *B. burgdorferi* is critical in the pathogenesis of this spirochete [58], and, thus, we were curious to further explore the interactions between r*Ixs*CRT and complement proteins. Published studies have reported that tick CRT could bind to C1q, one of the three subunits (C1q, C1, and C1s) of the C1 complex [28,32]. This study provides new insights into the functions of tick CRT by showing that r*Ixs*CRT binds to all three C1 complex complement proteins (C1q, C1r, and C1s) and to multiple intermediate complement proteins including C3, C5, C6, C7, C8, C9, complement factor I, and complement factor H. Further, our pull-down and ELISA analysis data demonstrated that although r*Ixs*CRT directly binds C1 complex proteins and intermediate and late complement proteins, it does not bind the activated C5b-9 or the membrane attack complex (MAC).

Complement cascades are activated via three different pathways: classical, alternative, and mannose-binding lectin (MBL). Activation of the classical complement pathway starts with C1q binding onto the microbe surface of antigens, which, in turn, activates two subunits of the C1 complex (C1r and C1s) [29,59]. Likewise, the MBL pathway is activated by the binding of MBL and ficolins to microbial surface oligosaccharides on microbe surfaces [60]. Lastly, the alternative pathway is activated by spontaneous hydrolysis of the internal C3 thioester bond and the binding of properdin to proteins, lipids, and carbohydrates on microbe surfaces [61]. The activation of the three pathways culminates in the activation of the C3 convertase, which leads to the C5 convertase, which reacts with complement proteins C6, C7, C8, and C9 to form C5b-9 or MAC. It is notable that despite the binding of the C1 complex proteins C3, C5, and C9, r*Ixs*CRT moderately blocked MAC deposition via the MBL pathway but not via the classical and alternative pathways. In our study, we noticed an increase in MAC deposition in the presence of r*Ixs*CRT in the classical and alternate pathways, which is consistent with our previous study, which showed that increasing amounts of recombinant *A. americanum* tick CRT led to the deposition of high amounts of MAC [32]. The fact that tick CRT binds complement proteins but moderately affects MAC deposition via the MBL pathway but not via the two other pathways is interesting and warrants additional investigations. The complement-sensitive strain of *B. burgdorferi* (B314/pBBE22luc) has been utilized to study inhibitors of the complement system in *B. burgdorferi* pathogenesis [38,62]. Our findings that r*Ixs*CRT did not rescue the complement-sensitive strain from complement killing may be explained by the fact that r*Ixs*CRT had a modest impact on the MBL pathway and no effect on the classical and alternate pathways. We speculate that the protective effects of *Ixs*CRT against the complement killing of *B. burgdorferi* were nullified by its lack of effect against classical and alternative activation pathways of complement, despite reducing MAC deposition via the MBL pathway.

We have previously shown that *B. burgdorferi*-infected *I. scapularis* nymphs highly secrete *Ixs*CRT during the 48–72 feeding time points, aligning with high *B. burgdorferi* transmission events after the tick has fed for more than 48 h [17,19]. This suggested that *B. burgdorferi* might directly interact with *Ixs*CRT to promote its colonization of the host. It is interesting to note that supplementing the BSK-II media with r*Ixs*CRT enhanced the growth pattern of *B. burgdorferi* under in vitro conditions. We speculate that exposure to *Ixs*CRT triggers the rapid growth of *B. burgdorferi* at the site of its transmission, thus facilitating the establishment of local dermal infection that precedes the colonization of secondary organs [63].

This study adds to our knowledge and raises questions warranting further investigations on the role of tick CRT in feeding and the transmission of *B. burgdorferi* by ticks. It is particularly interesting that although *Ixs*CRT bound the C1 complex complement proteins, it did not block the deposition of the MAC via the classical pathway. We speculate that in the complex environment of host skin, the *Ixs*CRT protein may serve as a decoy protein, potentially facilitating pathogen transmission by deviating the activation surface of complement from *B. burgdorferi* to *Ixs*CRT. Our DiffPOP data revealed that r*Ixs*CRT interacted with extracellular matrix proteins such as collagen. It is possible that at the tick-feeding site, native CRT bound to exposed extracellular matrix proteins could serve as the site of complement activation and in the process deviating complement attack of transmitted *B. burgdorferi*. However, this diversion of complement activation might also have implications for the tick. Further investigation is needed to fully understand the role of *Ixs*CRT in tick–pathogen interactions and its potential effects on tick physiology and immune evasion strategies. Moreover, it is also important to note that while the sequence of the r*Ixs*CRT produced in our study was confirmed to match the cDNA sequence available in GenBank, some properties of the recombinant protein may have been influenced by post-translational modifications in the yeast expression system, and it is likely that these modifications differed from those of the native tick protein. Therefore, the results presented in this study specifically pertain to r*Ixs*CRT, and further studies are required to substantiate these findings with the native tick calreticulin.

## Figures and Tables

**Figure 1 pathogens-13-00560-f001:**
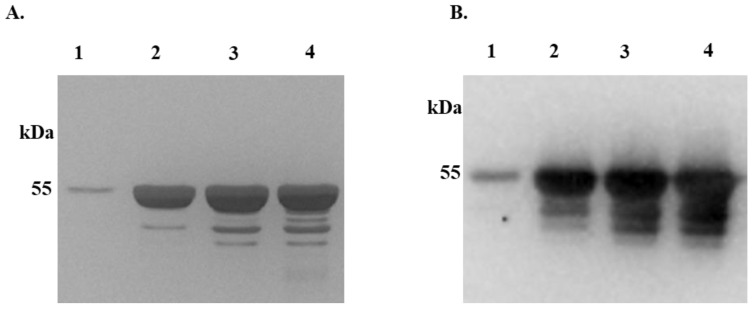
**Expression and affinity purification of recombinant *I. scapularis* calreticulin.** Recombinant *I. scapularis* calreticulin (r*Ixs*CRT, 47.61 kDa) with His-tag was expressed in *Pichia pastoris* over 3 days and affinity-purified under native conditions. Affinity purification of r*Ixs*CRT was validated by standard SDS-PAGE followed by silver staining (**A**) and Western blotting using the antibody for the Histidine fusion tag (**B**). Lanes 1–4 represent the 50, 75, 100, and 200 imidazole concentrations (mM) at which the r*Ixs*CRT was eluted, dialyzed in PBS, concentrated, and used for the assays.

**Figure 2 pathogens-13-00560-f002:**
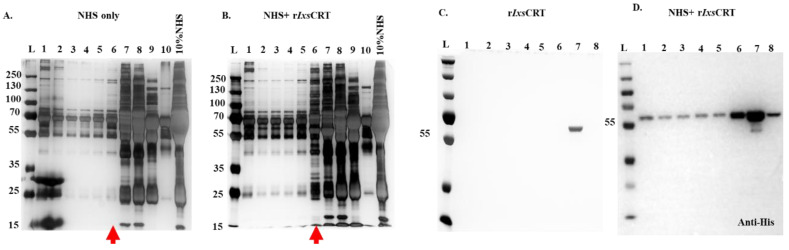
**Differential Precipitation of Proteins (DiffPOP) reveals interactions between r*Ixs*CRT and complement proteins.** Affinity-purified r*Ixs*CRT (10 µg) was pre-incubated with 10% normal human serum (NHS) for 90 min at 37 °C, while NHS and r*Ixs*CRT served as controls. The reactions were then stabilized and subjected to differential precipitation using escalating amounts of precipitating buffer, as described in the materials and methods section (see Appendix A). Different fractions obtained from the precipitation process were analyzed by standard SDS-PAGE and silver staining. (**A**) shows the protein profile of NHS only, (**B**) illustrates the profile of the NHS + r*Ixs*CRT mixture, and (**C**) displays the profile of r*Ixs*CRT only. Additionally, the NHS + r*Ixs*CRT mixture was subjected to Western blotting analysis using an antibody against the histidine fusion tag to track the precipitation of r*Ixs*CRT (**D**). The distinct protein profile between NHS and NHS + r*Ixs*CRT, highlighted by the red arrowhead in panels (**A**,**B**), indicated potential interactions between r*Ixs*CRT and specific proteins. The precipitation profile depicted in (**D**) guided the selection of fractions for subsequent LCMS/MS analysis, shedding light on the specific complement proteins interacting with r*Ixs*CRT. L: ladder depicting molecular weight (kDa), Lanes 1–10 = DiffPOP fractions 1–10.

**Figure 3 pathogens-13-00560-f003:**
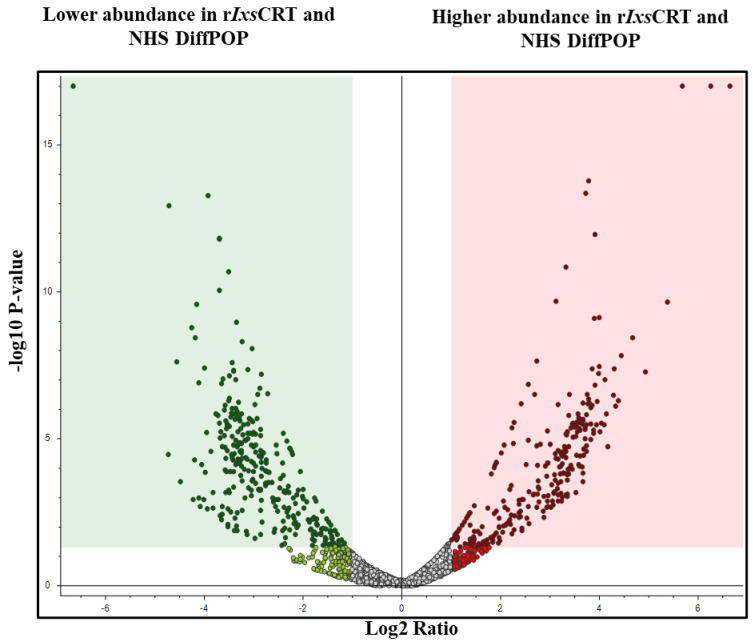
**Volcano plot analyses revealed human serum proteins that differentially co-precipitated with r*Ixs*CRT.** Normalized abundance values of three biological replicates were used in Proteome Discoverer™ 2.4 software (Thermo Fisher Scientific, Dallas, TX, USA) to generate the volcano plot. In the volcano plot, the Y-axis shows the −log10 *p*-value and the X-axis shows the magnitude of change (log2 fold change). Red dots represent proteins that co-precipitated in high amounts with r*Ixs*CRT, while green dots represent those that were absent or co-precipitated in low amounts in the presence of r*Ixs*CRT. An adjusted *p*-value ≤ 0.05 and log2 fold change of more than 2 were used as cut-offs to select proteins that co-precipitated in high amounts with r*Ixs*CRT.

**Figure 4 pathogens-13-00560-f004:**
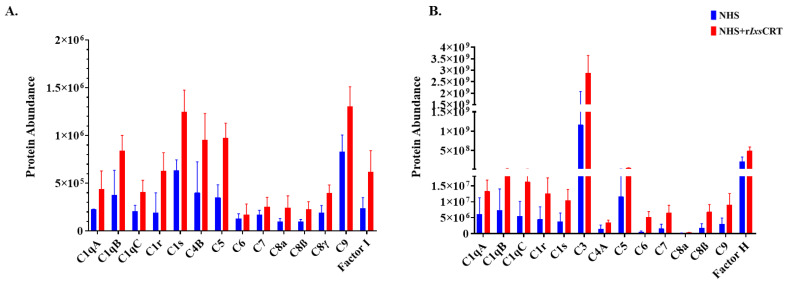
**Differential precipitation of complement proteins in the presence of r*Ixs*CRT.** Fractions obtained from the differential precipitation (DiffPOP) were pooled into group A (fractions 1–5), group B (fraction 6), and group C (fractions 7 and 8). These fractions were subjected to Liquid Chromatography-Mass Spectrometry (LCMS/MS) analysis to investigate the abundance of complement proteins interacting with r*Ixs*CRT. (**A**) illustrates the abundance of complement proteins interacting with r*Ixs*CRT in group A, while (**B**) depicts the corresponding interactions in group B. The Y-axis in (**A**,**B**) illustrates the protein abundance, indicating the levels of proteins co-precipitated with r*Ixs*CRT (depicted by the red graph) in comparison to NHS only (represented by the blue graph).

**Figure 5 pathogens-13-00560-f005:**
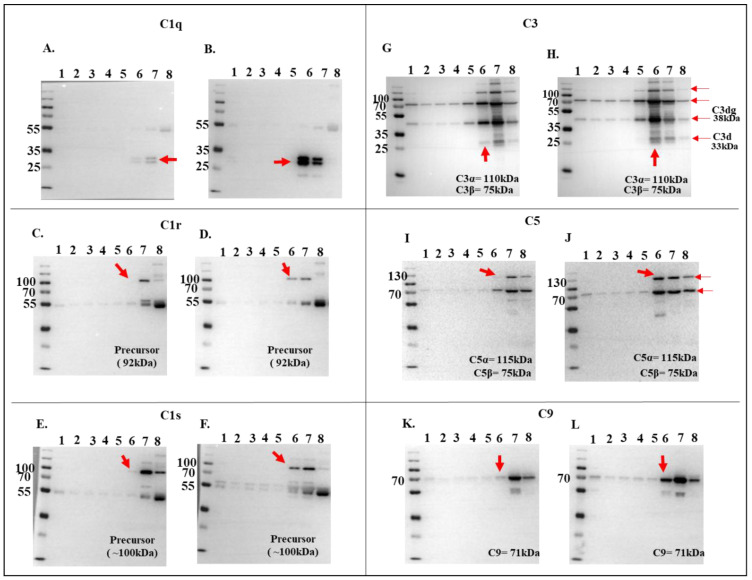
**Validation of complement protein and r*Ixs*CRT interactions via Western blotting analysis.** To confirm the interactions observed in LCMS/MS analysis (referenced in Figure 3), DiffPOP fractions were subjected to standard Western blotting using antibodies specific for complement proteins: C1q, C1r, C1s, C3, C5, and C9, as indicated. Distinctive binding patterns, highlighting differences between NHS + r*Ixs*CRT and the NHS control, are denoted by red arrowheads. Panels (**A**,**C**,**E**,**G**,**I**,**K**) depict immunoblots of NHS-only samples, while panels (**B**,**D**,**F**,**H**,**J**,**L**) represent immunoblots of NHS + r*Ixs*CRT samples.

**Figure 6 pathogens-13-00560-f006:**
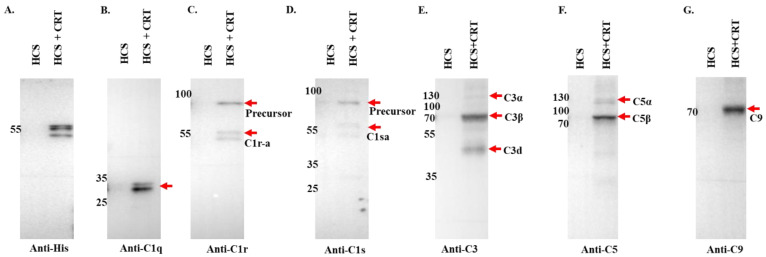
**Pull-down assay validated the r*Ixs*CRT binding of complement proteins.** Affinity-purified r*Ixs*CRT bound to His specific magnetic beads (Dynabeads™, Thermo Fisher Scientific, Waltham, MA, USA) was used to pull down complement proteins from human complement serum (HCS). The beads were washed and the eluted protein complexes were subjected to Western blotting analysis using antibodies for the histidine tag (**A**) and complement proteins C1q (**B**), C1r (**C**), C1s (**D**), C3 (**E**), C5 (**F**), and C9 (**G**). In all the panels (**A**–**G**), HCS is the human complement serum that was eluted from the empty beads, i.e., without the bait protein (r*Ixs*CRT), and HCS + r*Ixs*CRT denotes the human complement proteins that were pulled down and eluted from the beads loaded with r*Ixs*CRT. Red arrowheads denote the detected complement proteins at their expected molecular weight size (kDa).

**Figure 7 pathogens-13-00560-f007:**
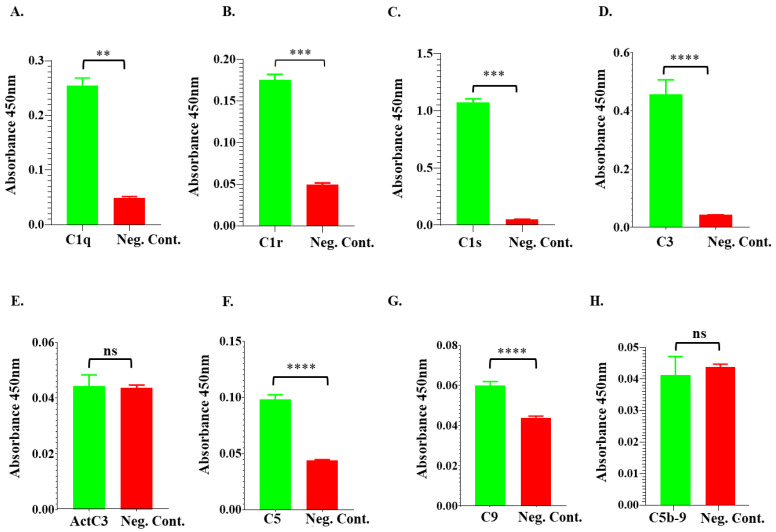
**ELISA analysis demonstrates the r*Ixs*CRT binding of complement proteins.** High binding ELISA plates coated with affinity-purified r*Ixs*CRT (250 ng) were incubated with normal human serum (NHS) followed by antibodies for C1q (**A**), C1r (**B**), C1s (**C**), C3 (**D**), activated C3 (**E**), C5 (**F**), C9 (**G**), and C5b-9 or MAC (**H**) antibodies. Y-axis denotes the absorbance measured at *A*_450nm_, which reflected the intensity of the specific antibody binding to r*Ixs*CRT. Non-coated wells blocked with 1% BSA were incubated with NHS and used as a negative control (Neg. Cont.). Data represent mean ± SEM of 3 biological replicates. For statistical analysis, *t*-test was performed on GraphPad Prism 9 and ** represents *p* < 0.01, *** represents *p* < 0.001, **** represents *p* < 0.0001, and ns represents not significant.

**Figure 8 pathogens-13-00560-f008:**
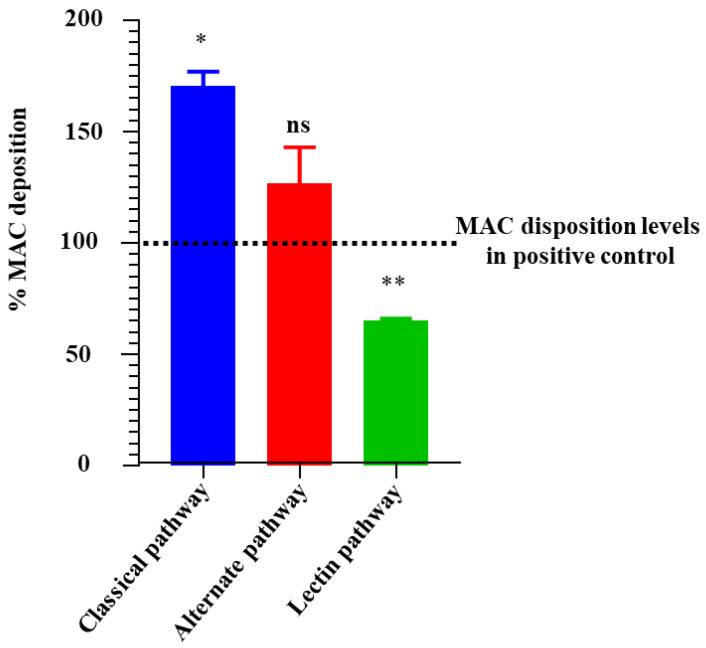
r*Ixs*CRT apparently enhances membrane attack complex (MAC) deposition in the classical pathway and alternate pathway but inhibits MAC deposition in the lectin pathway. The Wieslab^®^ Complement System Kit (Svar Life Science AB, Malmo, Sweden) was used to detect the effects of r*Ixs*CRT on MAC deposition via the classical pathway, alternative pathway, and lectin pathway, as described in the materials and methods section. In brief, r*Ixs*CRT (4 μM) was incubated with NHS (provided with the kit) at 37 °C for 30 min and then added to wells pre-coated with the antibody for MAC. Diluent and kit-provided reagent served as negative and positive controls, respectively. After washing, the conjugate and substrate were added according to the manufacturer’s instructions for each kit. NC denotes negative control and NHS denotes normal human serum, used as the positive control. % MAC deposition (Y-axis) was calculated as mentioned in the methodology section and the MAC deposition in the positive control was represented as 100% (denoted by the black dotted line). Data are presented as deposited MAC ± SEM calculated from 3 biological replicates. Statistical significance was determined by Student’s *t*-test in GraphPad Prism 9. * Represents *p* ≤ 0.05, ** represents *p* ≤ 0.01, and ns represents not significant.

**Figure 9 pathogens-13-00560-f009:**
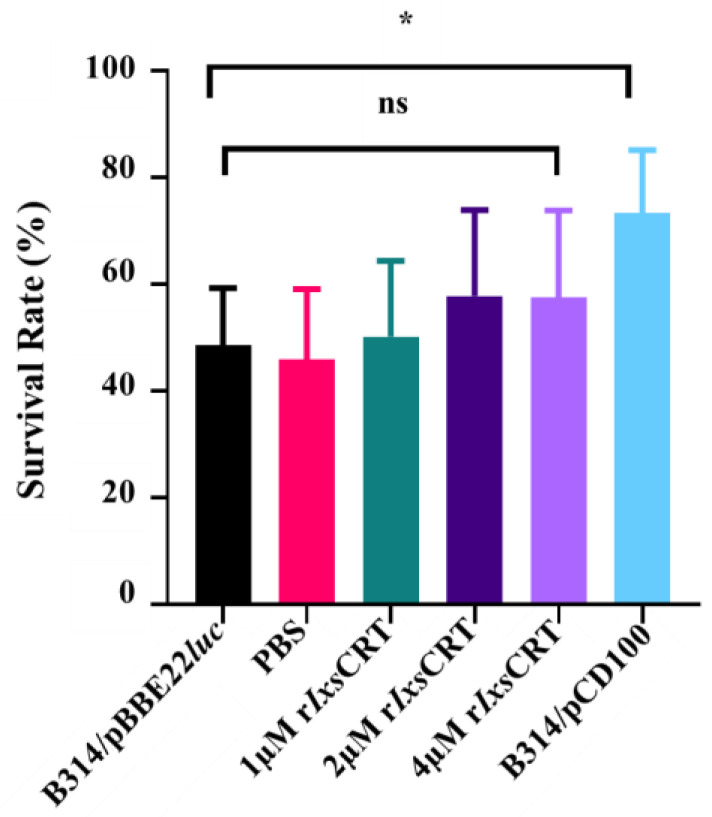
**r*Ixs*CRT does not protect *B. burgdorferi* from complement killing.** Normal human serum (NHS) was pre-incubated with serial dilutions of r*Ixs*CRT (1, 2, and 4 µM) or phosphate-buffered saline (PBS) at 37 °C for 30 min prior to the addition of 85 µL of 10^6^ cells/mL of *B. burgdorferi* B314/pBBE22luc (complement-sensitive strain) and incubated in a bio-shaker at 32 °C and 100 rpm. NHS incubated with *B. burgdorferi* B314/pPCD100 (complement-resistant strain) was used as a positive control. Survival rates of *B. burgdorferi* were assessed at 3 h post-incubation. Data represent mean ± SEM of 3 biological replicates. Statistical significance was evaluated using *t*-test in GraphPad Prism 9 (ns: no significance, * represents *p*-value ≤ 0.05).

**Figure 10 pathogens-13-00560-f010:**
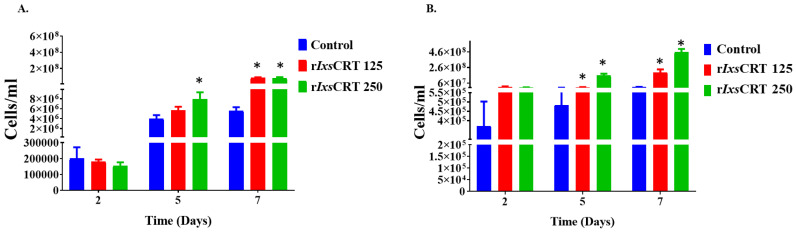
**r*Ixs*CRT promotes the growth of *B. burgdorferi* in culture.** *B. burgdorferi* (strain MSK5) cultured in the presence of r*Ixs*CRT at 2.6 mM (r*Ixs*CRT 125) and 5.2 mM (r*Ixs*CRT 250) were sampled at days 2, 5, and 7. (**A**) In triplicate, cells were quantified by manual counts on a Petroff-Hausser chamber and quantified using the following formula: number of cells/mL = Average of cells counted in all chambers × Dilution factor x 50,000. (**B**) *B. burgdorferi* was quantified by qPCR using the genomic DNA of the *B. burgdorferi* and *fLa*B primers. For ELISA, Student’s *t*-test was used for statistical analysis on GraphPad Prism 9 and *p*-value ≤ 0.05 (denoted by *) was considered significant for 3 biological replicates.

**Figure 11 pathogens-13-00560-f011:**
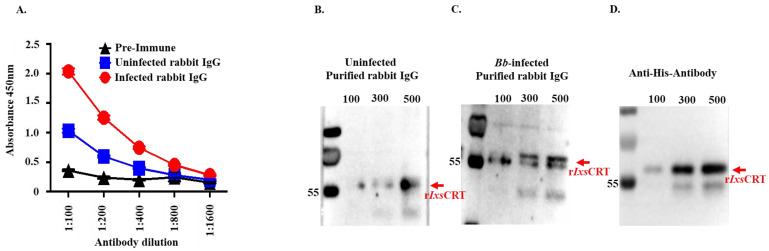
One-time exposure of rabbits to *B. burgdorferi*-infected *I. scapularis* nymphs triggers high IgG antibody levels for r*Ixs*CRT. (**A**) Affinity-purified r*Ixs*CRT (250 ng) was subjected to standard ELISA using serially diluted purified IgG of rabbits that were fed upon for a single time by uninfected (blue line graph) and *B. burgdorferi*-infected (red line graph) *I. scapularis* nymph ticks. The black line graph represents pre-immune IgG binding, which was used for a negative control. (**B**–**D**) Various quantities of affinity-purified r*Ixs*CRT (100, 300, and 500 ng) were subjected to Western blotting analysis using purified IgG from rabbits that were fed upon for a single time by uninfected (**B**) and *B. burgdorferi*-infected ticks (**C**) as well as the antibody for the histidine tag, which served as a positive control (**D**).

**Table 1 pathogens-13-00560-t001:** Reactome analysis depicting the pathways linked with proteins that were abundant ≥1.5-fold in NHS treated with CRT.

Pathway Name (Reactome Identifier)		Entities				Reactions	
		Found	Ratio	*p*-Value	FDR *	Found	Ratio
**Fractions 1–5**	Terminal pathway of complement (R-HSA-166665.4)	2/8	6.88 × 10^−4^	0.009	0.578	4/5	3.49 × 10^−4^
	Defective ABCA12 causes ARCI4B (R-HSA-5682294.3)	1/1	8.60 × 10^−5^	0.017	0.578	1/1	6.99 × 10^−5^
	Defective ABCC2 causes DJS (R-HSA-5679001.4)	1/1	8.60 × 10^−5^	0.017	0.578	1/1	6.99 × 10^−5^
	Regulation of complement cascade (R-HSA-977606.6)	6/135	0.012	0.03	0.578	20/42	0.003
	CD22-mediated BCR regulation (R-HSA-5690714.3)	4/70	0.006	0.033	0.578	3/4	2.80 × 10^−4^
	Recycling of bile acids and salts (R-HSA-159418.4)	2/18	0.002	0.039	0.578	2/17	0.001
	Complement cascade (R-HSA-166658.5)	6/146	0.013	0.041	0.578	33/72	0.005
	Defective factor XII causes hereditary angioedema (R-HSA-9657688.2)	1/3	2.58 × 10^−4^	0.05	0.578	2/2	1.40 × 10^−4^
	Defective SERPING1 causes hereditary angioedema (R-HSA-9657689.2)	1/3	2.58 × 10^−4^	0.05	0.578	1/3	2.10 × 10^−4^
**Fraction 6**	Terminal pathway of complement (R-HSA-166665.4)	4/8	6.88 × 10^−4^	9.00 × 10^−6^	0.004	5/5	3.49 × 10^−4^
	Regulation of complement cascade (R-HSA-977606.6)	9/135	0.012	3.14 × 10^−4^	0.064	13/42	0.003
	Complement cascade (R-HSA-166658.5)	9/146	0.013	5.50 × 10^−4^	0.075	22/72	0.005
	Response to elevated platelet cytosolic Ca^2+^ (R-HSA-76005.3)	7/133	0.011	0.005	0.429	2/14	9.79 × 10^−4^
	Regulation of Insulin-like Growth Factor (IGF) transport and uptake by Insulin-like Growth Factor Binding Proteins (IGFBPs) (R-HSA-381426.3)	6/124	0.011	0.014	0.61	2/14	9.79 × 10^−4^
	Platelet degranulation (R-HSA-114608.4)	7/128	0.011	0.004	0.429	2/11	7.69 × 10^−4^
	Dissolution of fibrin clot (R-HSA-75205.4)	2/13	0.001	0.018	0.61	15/21	0.001
	Post-translational protein phosphorylation (R-HSA-8957275.2)	5/107	0.009	0.027	0.61	1/1	6.99 × 10^−5^
	Other semaphoring interactions (R-HSA-416700.2)	2/19	0.002	0.036	0.61	3/9	6.29 × 10^−4^
	Sema4D-induced cell migration and growth-cone collapse (R-HSA-416572.4)	2/20	0.002	0.04	0.61	4/7	4.89 × 10^−4^
	RHO GTPases activate CIT (R-HSA-5625900.3)	2/20	0.002	0.04	0.61	3/6	4.19 × 10^−4^
	Common pathway of fibrin clot formation (R-HSA-140875.5)	2/22	0.002	0.047	0.61	6/29	0.002
**Fractions 7–8**	Collagen chain trimerization (R-HSA-8948216.4)	8/44	0.004	1.24 × 10^−4^	0.091	7/28	0.002
	Collagen degradation (R-HSA-1442490.4)	8/64	0.006	0.001	0.331	11/34	0.002
	Anchoring fibril formation (R-HSA-2214320.4)	4/15	0.001	0.002	0.331	4/4	2.80 × 10^−4^
	Collagen biosynthesis and modifying enzymes (R-HSA-1650814.5)	8/67	0.006	0.002	0.331	30/51	0.004
	Integrin cell–surface interactions (R-HSA-216083.5)	9/85	0.007	0.002	0.331	11/55	0.004
	TRKA activation by NGF (R-HSA-187042.2)	2/3	2.58 × 10^−4^	0.005	0.551	4/4	2.80 × 10^−4^
	Collagen formation (R-HSA-1474290.4)	8/90	0.008	0.011	0.763	39/77	0.005
	NCAM1 interactions (R-HSA-419037.2)	5/42	0.004	0.013	0.763	1/10	6.99 × 10^−4^
	ECM proteoglycans (R-HSA-3000178.5)	7/76	0.007	0.014	0.763	7/23	0.002
	Assembly of collagen fibrils and other multimeric structures (R-HSA-2022090.4)	6/61	0.005	0.016	0.763	9/26	0.002
	Laminin interactions (R-HSA-3000157.3)	4/30	0.003	0.018	0.763	4/15	0.001
	Degradation of the extracellular matrix (R-HSA-1474228.5)	10/140	0.012	0.019	0.763	13/105	0.007
	Crosslinking of collagen fibrils (R-HSA-2243919.4)	3/18	0.002	0.022	0.763	1/13	9.09 × 10^−4^
	Activation of TRKA receptors (R-HSA-187015.3)	2/7	6.02 × 10^−4^	0.023	0.763	8/8	5.59 × 10^−4^
	Defective SLC2A9 causes hypouricemia renal 2 (RHUC2) (R-HSA-5619047.3)	1/1	8.60 × 10^−5^	0.032	0.763	1/1	6.99 × 10^−5^
	Role of second messengers in netrin-1 signaling (R-HSA-418890.2)	2/10	8.60 × 10^−4^	0.043	0.763	1/4	2.80 × 10^−4^
	Signaling by PDGF (R-HSA-186797.5)	5/60	0.005	0.049	0.763	1/31	0.002

* FDR: false discovery rate.

## Data Availability

The original contributions presented in the study are included in the article/Appendix A, further inquiries can be directed to the corresponding author/s.

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
