# Peer review of "Recombinant *Ixodes scapularis* Calreticulin Binds Complement Proteins but Does Not Protect *Borrelia burgdorferi* from Complement Killing"

_pathogens, 2024, doi:10.3390/pathogens13070560_

Round 1
Reviewer 1 Report
Comments and Suggestions for Authors
It is a pleasure to review this manuscript, which is very well written and has very consistent hypotheses and experiments to answer this question of the role of calreticulin and the transmission of Borrelia burgdoferi or the colonization of the host.
Below is a minimal list of suggestions for corrections:
Materials and Methods
- Line 97: the font size should be corrected;
- Line 267 (2.9): The word “in” is repeated;
- Line 288: adjusted to the range of the standard curve, it is not necessary to describe the entire range, just 101 to 108;
The time expressed in minutes, sometimes the word “min” and sometimes “minutes”, please unify throughout the text;
Results
I would like to suggest remove the Tabl1 1 to a Supplementary files;
Fig 1.:Perhaps it would be more informative to indicate the size of the expected rIxs_CRT product.
Fig. 2B: Adjust the y axis.
Author Response
Responses to Reviewer 1:
Thank you for taking the time to review and give comments on our research article, entitled ‘Ixodes scapularis calreticulin binds complement proteins but does not protect Borrelia burgdorferi from complement killing’.
We have addressed the comments received, and our point-to-point responses can be found below
Materials and Methods
Comment: Line 97: the font size should be corrected.
Response: Thank you for pointing it out. The correction is incorporated in the manuscript.
Comment: Line 267 (2.9): The word “in” is repeated.
Response: Thank you for pointing it out. The correction is incorporated in the manuscript.
Comment: Line 288: adjusted to the range of the standard curve, it is not necessary to describe the entire range, just 101 to 108.
Response: Thank you for pointing it out. The correction is incorporated in the manuscript.
Comment: The time expressed in minutes, sometimes the word “min” and sometimes “minutes”, please unify throughout the text.
Response: Thank you for pointing it out. The correction is incorporated in the manuscript.
Results
Comment: I would like to suggest remove the Table1 1 to a Supplementary file.
Response: Thank you for your suggestion. We appreciate your feedback regarding the placement of Table 1 in the supplementary files. However, we believe that including Table 1 in the main manuscript provides readers with direct access to the pathways affected by rIxsCRT, offering valuable insights into our findings without the need to refer to supplementary materials. Additionally, the descriptive raw data used to generate Table 1 is available in the supplementary file, ensuring transparency and facilitating further analysis if desired.
Comment: Fig 1.: Perhaps it would be more informative to indicate the size of the expected rIxsCRT product.
Response: Thank you for pointing it out. The correction is incorporated in the manuscript in line 328.
Comment: Fig. 2B: Adjust the y axis.
Response: We are not able to make this adjustment because figure 2B is a silver stain of SDS gel.

Reviewer 2 Report
Comments and Suggestions for Authors
General comments:
1. Extensive studies are reported to identify and characterise the human plasma proteins to which tick calreticulin binds. The methods used are well established. Were alternative methodologies considered such as surface plasmon resonance spectroscopy that enable quantitative analyses of protein-protein interactions.
2. Apart from molecular weight, what is the evidence the recombinant protein replicates the binding properties of the native I. scapularis protein?
3. The title and conclusion state that Ixodes scapularis calreticulin does not protect Borrelia burgdorferi from complement-mediated killing. However, evidence for this statement is ‘not shown.’
4. The authors speculate that Ixodes scapularis calreticulin may act as a decoy activator of complement once secreted into the skin feeding site. Although such activation might benefit pathogen transmission and infection of the vertebrate host, what effect might it have on the tick?
Specific comments:
Line 49: ‘absence of effective vaccines against LD and other TBD agents’ overlooks the success of tick-borne encephalitis vaccines.
Lines 53-55: Citation of a 1939 publication does not support the claim that immunization against tick feeding ‘has emerged as the most promising alternative tick control method.’ A more recent publication is required although the claim is debatable especially for human use.
Lines 269-270: what is mean by ‘empirically determined’? Is the concentration known of tick calreticulin in tick saliva? A dose response curve should strengthen the results.
Lines 423-424: Suggest alternative wording: ‘Unique binding patterns distinguishing NHS + rIxsCRT from NHS control are indicated by red arrowheads.’
Line 428: ‘To definitively ascertain’ is poor grammar. It is questionable whether the technique is ‘definitive’ – does it distinguish between protein-protein and non-protein based interactions; does it distinguish intermediary/indirect binding?
Lines 478-480: Complement-mediated killing: the data need to be shown! Since this result is an important part of the manuscript, the results should be shown in the text (rather than as supplementary data). The data should be strengthened by considering alternative approaches/additional controls for testing complement-mediated killing and comparative testing with human, rabbit and murine serum.
Lines 483-484: Blood clotting: the data should be included in the supplementary material.
Line 510: what is the ‘indicated concentration.’ What about a dose response curve?
Line 518: IxsCRT not rIxsCRT was identified in tick saliva. What is the evidence rIxsCRT is the same as IxsCRT identified in the previous study?
Figure 10: Is the method of one-time exposure of rabbits to infected tick feeding described? Are rabbits natural hosts of Ixodes scapularis nymphs? What happens when nymphs feed on deer mice?
Comments on the Quality of English Language
Minor editing suggested.
Author Response
Response to Reviewer 2
Thank you for taking the time to review and give comments on our research article, entitled ‘Ixodes scapularis calreticulin binds complement proteins but does not protect Borrelia burgdorferi from complement killing’. We have addressed the comments received, and our point-to-point responses can be found below.
General comments:
Comment: Extensive studies are reported to identify and characterize the human plasma proteins to which tick calreticulin binds. The methods used are well established. Were alternative methodologies considered such as surface plasmon resonance spectroscopy that enable quantitative analyses of protein-protein interactions.
Response: Thank you for your insightful comment. We appreciate your suggestion regarding alternative methodologies such as surface plasmon resonance spectroscopy for quantitative analyses of protein-protein interactions. While our current study utilized three independent methods (DiffPoP assay, pull-down assay, and LC-MS/MS) to support our findings, we acknowledge the value of incorporating additional techniques for a more comprehensive analysis in future studies. Thank you for highlighting this avenue for improvement, and we will ensure to address it in our further investigations.
Comment: Apart from molecular weight, what is the evidence the recombinant protein replicates the binding properties of the native I. scapularis protein?
Response: The reviewer has brought up a valuable point worth following up on. However, the goal of this study was to utilize the recombinant protein to understand functions of this important protein. We acknowledge that the functional efficiency of the native protein may be superior to the recombinant protein and it will be worthwhile to attempt isolating native calreticulin.
Comment: The title and conclusion state that Ixodes scapularis calreticulin does not protect Borrelia burgdorferi from complement-mediated killing. However, evidence for this statement is ‘not shown.’
Response: The data for the complement killing assay is now included in the supplementary file as Supplementary File S6.
Comment: Reviewer comment: The authors speculate that Ixodes scapularis calreticulin may act as a decoy activator of complement once secreted into the skin feeding site. Although such activation might benefit pathogen transmission and infection of the vertebrate host, what effect might it have on the tick?
Response: We appreciate your insightful comment. In our study, we propose that Ixodes scapularis calreticulin (IxsCRT) may serve as a decoy activator of the complement system once secreted into the skin feeding site during tick bites. While such complement activation could potentially benefit Borrelia transmission and infection of the vertebrate host, it's also important to consider the potential effects on the tick itself. We suggest that the complement system might become activated on surfaces coated with tick calreticulin, rather than on the Borrelia itself.
We hypothesize that by binding to IxsCRT, the complement system may be diverted away from attacking Borrelia, potentially facilitating pathogen transmission. However, this diversion of complement activation might also have implications for the tick.
Further investigation is needed to fully understand the role of IxsCRT in tick-pathogen interactions and its potential effects on tick physiology and immune evasion strategies. We appreciate your thoughtful consideration of these implications and will explore them further in our future research endeavors.
The important points related to this comment is now included in the manuscript at line 625-626 and line 631-634.
Specific comments:
Comment: Line 49: ‘absence of effective vaccines against LD and other TBD agents’ overlooks the success of tick-borne encephalitis vaccines.
Response: Thank you for pointing it out. Necessary changes have been incorporated in this section for better understanding.
Comment: Lines 53-55: Citation of a 1939 publication does not support the claim that immunization against tick feeding ‘has emerged as the most promising alternative tick control method.’ A more recent publication is required although the claim is debatable especially for human use.
Response: Thank you for raising this point. The important references are added to this section.
Comment: Lines 269-270: What is mean by ‘empirically determined’? Is the concentration known of tick calreticulin in tick saliva? A dose response curve should strengthen the results.
Response: The concentration of CRT in the tick saliva is still not determined. The word ‘empirically determined’ is removed from the sentence to avoid any further confusion.
Comment: Lines 423-424: Suggest alternative wording: ‘Unique binding patterns distinguishing NHS + rIxsCRT from NHS control are indicated by red arrowheads.’
Response: Thank you for the suggestion the sentence is now modified to ‘Distinctive binding patterns, highlighting differences between NHS + rIxsCRT and NHS control, are denoted by red arrowheads.’
Comment: Line 428: ‘To definitively ascertain’ is poor grammar. It is questionable whether the technique is ‘definitive’ – does it distinguish between protein-protein and non-protein based interactions; does it distinguish intermediary/indirect binding?
Response: Thank you for pointing out. The word ‘definitively ascertain’ is removed from this section.
Comment: Lines 478-480: Complement-mediated killing: the data need to be shown! Since this result is an important part of the manuscript, the results should be shown in the text (rather than as supplementary data). The data should be strengthened by considering alternative approaches/additional controls for testing complement-mediated killing and comparative testing with human, rabbit and murine serum.
Response: Thank you for your feedback and valuable suggestions. We appreciate your acknowledgment of the importance of the complement-mediated killing results. However, we believe that presenting this data as supplementary information allows interested readers to access and evaluate the results while maintaining the clarity and focus of the main text. The complement sensitivity data is available in supplemental table S6. Moreover, while comparative testing with human, rabbit, and murine serum could indeed provide valuable insights, and we believe that this aspect warrants further investigation in future research endeavors dedicated specifically to comparative serum testing.
Comment: Lines 483-484: Blood clotting: the data should be included in the supplementary material.
Response: Thank you for the suggestion. The data is now included as supplementary figure S1.
Line 510: What is the ‘indicated concentration.’ What about a dose response curve?
Response: The ‘indicated concentration’ is replaced with the actual protein concentration (2.6mM) which is the concentration of rIxsCRT used for the assay.
Comment: Line 518: IxsCRT not rIxsCRT was identified in tick saliva. What is the evidence rIxsCRT is the same as IxsCRT identified in the previous study?
Response: We are confident that rIxsCRT and native IxsCRT have the same primary sequence as the recombinant protein expression construct was based IxsCRT cDNA sequence in GenBank and verified by sequencing.
Comment: Figure 10: Is the method of one-time exposure of rabbits to infected tick feeding described? Are rabbits natural hosts of Ixodes scapularis nymphs? What happens when nymphs feed on deer mice?
Response: These data were published in Kim TK, Tirloni L, Bencosme-Cuevas E, Kim TH, Diedrich JK, Yates JR 3rd, Mulenga A. Borrelia burgdorferi infection modifies protein content in saliva of Ixodes scapularis nymphs. BMC Genomics. 2021 Mar 4;22(1):152. doi: 10.1186/s12864-021-07429-0. PMID: 33663385; PMCID: PMC7930271. doi: 10.1186/s12864-021-07429-0. The reference is now included in the manuscript under methodology section.

Round 2
Reviewer 2 Report
Comments and Suggestions for Authors
Please address the following comments:
1) Given the authors have not demonstrated that the
recombinant protein replicates the binding properties of the native I. scapularis protein, the manuscript needs to qualify the claims. For example, the title should be: “Recombinant Ixodes scapularis calreticulin binds complement…”. Although the sequence of the recombinant protein was confirmed as having the same primary sequence as the cDNA sequence in GenBank, it is possible/likely binding properties are influenced by post-translational modifications of the protein. As a yeast expression system was used to produce the recombinant (rather than a eukaryotic system e.g. baculovirus/insect cells), post-translational modifications of the recombinant protein will most likely differ from those of the native tick protein. The text should make clear the results relate to the recombinant protein and further studies are needed to show they reflect properties of the native tick calreticulin.
Lines 49-51: mention the TBEV vaccine
Line 269: delete “empirically determined”
Line 519: The native protein was identified as the tick saliva protein not the recombinant protein. Change rIxsCRT to IxsCRT. However, the authors should note the cDNA sequence obtained from GenBank has not been confirmed as identical to the protein they observed in their previous studies.
Supplemental table S6: The authors claim rIxsCRT does not protect B. burgdorferi from complement killing. Evidence supporting this claim must be shown up front and not in the supplementary information. The evidence is not supplementary to the paper – it is a main conclusion. What was the concentration of rIxsCRT with pCD100? The results should include standard errors and statistical analysis.
Figure 9: a dose response curve is needed to support the conclusion rIxsCRT promotes growth.
Figure 10: Please comment on the fact deer mice rather than rabbits are the common host of I. scapularis nymphs. Have any comparable data been published to show one time-exposure of natural hosts to infected and uninfected nymphs gives similar results?
Fig. S1: Indicate the concentration of rIxsCRT. A dose response curve is needed to support the conclusion of no effect.
Comments on the Quality of English Language
Some of the changes made to the text need editing.
Author Response
Reviewer 2 Comments and responses:
1) Comment:
a. Given the authors have not demonstrated that the recombinant protein replicates the binding properties of the native I. scapularis protein, the manuscript needs to qualify the claims. For example, the title should be: “Recombinant Ixodes scapularis calreticulin binds complement…”.
b. Although the sequence of the recombinant protein was confirmed as having the same primary sequence as the cDNA sequence in GenBank, it is possible/likely binding properties are influenced by post-translational modifications of the protein. As a yeast expression system was used to produce the recombinant (rather than a eukaryotic system e.g. baculovirus/insect cells), post-translational modifications of the recombinant protein will most likely differ from those of the native tick protein. The text should make clear the results relate to the recombinant protein and further studies are needed to show they reflect properties of the native tick calreticulin.
Response: Thank you for your thoughtful comments and suggestions. We have addressed both of your points as follows:a. We have incorporated the changes into the manuscript. The title has been revised to: “Recombinant Ixodes scapularis calreticulin binds complement…”.b. We acknowledge the potential differences in post-translational modifications between the recombinant protein produced in a yeast expression system and the native I. scapularis protein. While we confirmed that the primary sequence of the recombinant protein matches the cDNA sequence in GenBank, we agree that post-translational modifications could influence binding properties. We have clarified in the text (line 678- line 689) that our results pertain specifically to the recombinant protein. Additionally, we have noted that further studies are necessary to determine whether these findings accurately reflect the properties of the native tick calreticulin.
2) Comment: Lines 49-51: mention the TBEV vaccine.
Response: Thank you for the suggestion. The sentence is now modified in line 50-54 as ‘In the absence of effective vaccines against Lyme disease (LD), killing ticks using acaricides, personal protection measures, and avoiding infected ticks are the only available options to prevent LD. However, it is worth noting that a vaccine is available for tick-borne encephalitis virus (TBEV), highlighting the potential for vaccine development against other tick-borne diseases.’
3) Line 269: delete “empirically determined”
Response: Thank you for your suggestion. We have already deleted "empirically determined" during the revisions made in Round 1.
4) Comment: Line 519: The native protein was identified as the tick saliva protein not the recombinant protein. Change rIxsCRT to IxsCRT. However, the authors should note the cDNA sequence obtained from GenBank has not been confirmed as identical to the protein they observed in their previous studies.
Response: Thank you for the suggestions. The changes are incorporated in the manuscript.
5) Supplemental table S6: The authors claim rIxsCRT does not protect B. burgdorferi from complement killing. Evidence supporting this claim must be shown up front and not in the supplementary information. The evidence is not supplementary to the paper – it is a main conclusion. What was the concentration of rIxsCRT with pCD100? The results should include standard errors and statistical analysis.
Response: The figure S6 with the statistical analysis is now updated and added in main manuscript as Figure 9.
6) Figure 9: a dose response curve is needed to support the conclusion rIxsCRT promotes growth.
Response: This now Figure 10A and B with the dose response graph using 2.6mM and 5.2mM rIxsCRT 250.
7) Figure 10: Please comment on the fact deer mice rather than rabbits are the common host of I. scapularis nymphs. Have any comparable data been published to show one time-exposure of natural hosts to infected and uninfected nymphs gives similar results?
Response: Thank you for your insightful question. To the best of our knowledge, there have been no published studies directly comparing the results of one-time exposure of natural hosts (such as deer mice) to infected and uninfected nymphs under similar conditions. However, the use of rabbits as hosts in tick studies is well-documented and provides a controlled environment that allows for reliable and reproducible results. This established methodology is crucial for ensuring the consistency and validity of our findings.While it is true that deer mice are common hosts for I. scapularis nymphs in the wild, our study utilized rabbits as hosts for several reasons. Primarily, there is a substantial body of previous research in our lab using rabbits for tick studies, providing a strong comparative basis for our results. Using rabbits allowed us to ensure consistency and reliability in our experimental conditions, which is crucial for the validity of our findings. We acknowledge that further studies involving deer mice would be valuable to fully understand the interactions in a natural setting.Additionally, please refer to Supplementary File SF2A and SF2B in the publication (https://www.ncbi.nlm.nih.gov/pmc/articles/PMC7930271/) for the details of purified Ab97 (Bb infected) and Ab98 (Bb uninfected) antibodies obtained after a single infestation on rabbits. It is also mentioned in the manuscript as reference 17.
8) Fig. S1: Indicate the concentration of rIxsCRT. A dose response curve is needed to support the conclusion of no effect.
Response: The figure S1 is now updated with dose response.
